# Nutrition in Acute Pancreatitis: From the Old Paradigm to the New Evidence

**DOI:** 10.3390/nu15081939

**Published:** 2023-04-18

**Authors:** Sara Sofia De Lucia, Marcello Candelli, Giorgia Polito, Rossella Maresca, Teresa Mezza, Tommaso Schepis, Antonio Pellegrino, Lorenzo Zileri Dal Verme, Alberto Nicoletti, Francesco Franceschi, Antonio Gasbarrini, Enrico Celestino Nista

**Affiliations:** 1Department of Medical and Surgical Sciences, Università Cattolica del Sacro Cuore, Fondazione Policlinico Universitario A. Gemelli IRCSS, 00168 Rome, Italy; 2Department of Emergency, Anesthesiological and Reanimation Sciences, Università Cattolica del Sacro Cuore, Fondazione Policlinico Universitario A. Gemelli IRCSS, 00168 Rome, Italy

**Keywords:** acute pancreatitis, nutrition, immunonutrition, probiotic

## Abstract

The nutritional management of acute pancreatitis (AP) patients has widely changed over time. The “pancreatic rest” was the cornerstone of the old paradigm, and nutritional support was not even included in AP management. Traditional management of AP was based on intestinal rest, with or without complete parenteral feeding. Recently, evidence-based data underlined the superiority of early oral or enteral feeding with significantly decreased multiple-organ failure, systemic infections, surgery need, and mortality rate. Despite the current recommendations, experts still debate the best route for enteral nutritional support and the best enteral formula. The aim of this work is to collect and analyze evidence over the nutritional aspects of AP management to investigate its impact. Moreover, the role of immunonutrition and probiotics in modulating inflammatory response and gut dysbiosis during AP was extensively studied. However, we have no significant data for their use in clinical practice. This is the first work to move beyond the mere opposition between the old and the new paradigm, including an analysis of several topics still under debate in order to provide a comprehensive overview of nutritional management of AP.

## 1. Introduction

Nutrition plays a decisive role in the prevention and care of different medical disorders with significant health impacts [1,2]. This awareness has developed over the years, making nutrition rightfully included in the guidelines for several diseases. History provides multiple examples, with AP being perhaps the clearest and most complete one [2]. Traditionally, fasting was considered necessary to obtain “pancreatic rest”. The idea behind this concept is that reduced stimuli for the secretion of protein enzymes is the main factor in hampering pancreatic inflammatory activity. This is strengthened by the local and nonspecific nature of AP inflammatory reaction during its first phases resulting from hydrolytic enzymes secretion as well as toxins and cytokines secretions [3]. Later, the attention moved to what could be defined as “the fasting problem”. AP patients should be evaluated as exhibiting moderate to elevated nutritional risk [2]. In fact, specific and nonspecific metabolic alterations follow AP, leading to an hypermetabolism and hypercatabolism state with a negative nitrogen balance [4]. Therefore, prevention and treatment of malnutrition with appropriate nutritional support should have a positive influence on the course of the disease and help avoid its complications [5]. Moreover, prolonged enteral starvation is linked to a series of mechanisms resulting in a condition called “leaky gut”, which is a further mechanism contributing to increased systemic inflammation, sepsis, multiple organ failure (MOF), and death [6]. Questions regarding the choice between enteral or parenteral nutrition, the optimal timing, route, and formula for enteral nutrition arises from these findings and has led to a paradigm shift of the AP nutritional approach. The central role of nutritional support in AP has brought out a whole series of considerations regarding the function of enteral and parenteral nutrition, and the ideal timing and method of administering enteral nutrition in conjunction with the optimal enteral formula. The aim of this work is to collect and analyze evidence of the nutritional aspects of AP management to investigate its impact. For its unprecedented methodology, this work provides a new perspective of AP management, simultaneously focusing on timing, nutrition route, and formula.

## 2. Materials and Methods

The present review displays an insight into the nutritional implications of AP. Relevant publications were extracted from PubMed, Scopus, and EMBASE databases, exploiting the following keywords: “acute pancreatitis”, “mild acute pancreatitis”, “severe acute pancreatitis”, “nutrition”, “enteral feeding”, “enteral nutrition”, “nasogastric tube”, “nasojejunal tube”, “parenteral nutrition”, “immunonutrition”, “immunomodulating nutrition”, “prebiotic in acute pancreatitis”, and “probiotic in acute pancreatitis”.

## 3. Acute Pancreatitis

AP is an inflammatory pancreatic disorder that is directly linked to a significant risk of morbidity and death [7]. The annual growth rate of AP incidence has reached 3.07%, showing a constant increase over the years [8]. Xiao et al. calculated in their systematic review that the worldwide incidence of AP (34/100,000 cases per year) is almost twice that of chronic pancreatitis and pancreatic cancer, with no significant differences between men and women [9]. The increasing incidence trend of acute pancreatitis may be correlated with the greater availability of diagnostic tests for AP, but also with the higher incidence of metabolic syndrome and obesity, both of which have been linked to gallstone disease and alcohol consumption [10,11]. Indeed, biliary AP represents, globally, the main cause of pancreatitis (45%), which has a double incidence compared to alcohol-induced AP (20%) [12]. Other rare causes of pancreatitis include autoimmune, hypertriglyceridemia, post endoscopic retrograde cholangiopancreatography (ERCP), hypercalcemia, malformative (pancreas divisum), neoplastic, genetic, and drug-induced forms [13,14]. Up to 15–25% of patients receive diagnosis of idiopathic pancreatitis after standard diagnostic tests [12,15]. The Revised Atlanta Classification stated that the diagnosis of AP is confirmed if at least two of the following three criteria are met: typical abdominal pain and/or rise in serum amylase or lipase over three times the upper limit of normal (ULN) and/or findings on abdominal imaging specific for AP [16]. AP may be divided into three groups, depending on the severity of the disease: mild, moderately severe, and severe [16]. AP is defined as mild if organ failure and local and systemic complications do not occur. Instead, temporary organ failure, local complications, or aggravation of concomitant diseases define moderately severe forms of AP. Finally, severe forms of AP are defined by an organ failure of more than 48 h and have been associated with a 30–40% mortality rate [17]. Mild AP is the most frequent form, and it presents a self-limiting course with a very low mortality (1%) [17]. Two overlapping phases (early and late) may be identified during an episode of AP [16]. The early phase is the result of local pancreatic inflammation, lasting about seven days, and leads to diffuse immune disturbances. The pancreatic injury initiates a cytokine storm which leads to systemic inflammatory response syndrome (SIRS). The length of SIRS may head towards multiorgan dysfunction syndrome (MODS), with the severity of AP being determined by the extent of time MODS is experienced, while the late phase may exhibit an extended duration of several weeks or months, and is distinguished by persistence of systemic signs of inflammation and/or local complication [16]. Different scoring systems exist to help identify patients at risk of developing a severe form of AP, in order to guide management and improve outcomes. The bedside index of severity in acute pancreatitis (BISAP) score is considered one of the most feasible scoring systems in everyday clinical practice, as it takes into consideration clinical data during the 24 h timeframe from admission [18]. It is considered accurate in predicting severity, organ failure, and death. The BISAP score takes into consideration five variables: blood urea nitrogen level > 25 mg/dL, impaired mental status, occurrence of systemic inflammatory response syndrome (SIRS), age exceeding 60 years, and existence of pleural effusion. Each of the following criteria scores one point. Individuals who present a BISAP score equal to or greater than 3 have a higher risk of mortality [19]. The cornerstones of managing AP entail administering intensive intravenous hydration, pain management, and appropriate nutritional support.

## 4. Clinical Nutrition

Nutrition plays a vital role in medicine [20] and should be considered as an integral element of patient care [1]. The literature agrees on nutrition being responsible for chronic diseases, both in terms of excess and/or lack of energy and nutrients [21,22]. Some of these, including obesity, cardiovascular disease, diabetes mellitus, and some types of cancer, are among the main causes of disability and death [22]. A suboptimal diet is responsible for one of every five deaths across the globe, outnumbering any other risk factor inclusive of smoking habit [23]. This evidence points to the potential for food and nutrition interventions to come to the fore as a major determinant in the prevention, management, and care of chronic conditions [22,24]. Moreover, acute and chronic diseases themselves affect nutrition in terms of food intake and metabolism, resulting in nutrition-related conditions [20]. Malnutrition, also known as undernutrition, is defined as a condition coming from lack of nutrition intake or uptake resulting in modified body composition and body cell mass as well as reduced physical and mental capability and impaired clinical outcome from disease [20,25,26]. According to the ESPEN (European Society for Clinical Nutrition and Metabolism) criteria [27], malnutrition diagnosis should be centered on either a low body mass index (BMI < 18.5 kg/m^2^) or on the combined finding of unintentional slimming (mandatory) and at least one of either reduced BMI (age-specific) or a low FFMI (fat-free body mass index). According to the ASPEN (American Society for Parenteral and Enteral Nutrition) and Academy Criteria [28], malnutrition can be diagnosed by the presence of a minimum of two out of the six standardized characteristics which reflect a patient’s nutrition status. Subordinately to the general diagnosis, malnutrition can be disease-related (DRM, disease-related malnutrition) or disease-free. The former is additionally characterized by the presence of inflammation which can be chronic (cachexia), acute (injury related), or absent [20]. The malnutrition’s etiology is complex and multifactorial as it includes organic, mental, and social risk factors [29]. The single most important factor causing malnourishment is low dietary consumption, typically resulting from a reduction in appetite sensation [25] and/or the inability to chew or swallow [30]. Macro- and micronutrient malabsorption are other main factors exemplary in patients undergoing abdominal surgery or suffering from gastrointestinal diseases [25]. Lastly, increased losses or increased energy expenditure also result in malnutrition, such as that occurring in patients with trauma or burns [25,31]. A large survey conducted by the British Association of Parenteral and Enteral Nutrition (BAPEN) in 2021 [32] reported that malnourishment prevalence was highest in individuals with gastrointestinal disorders (48%), lung diseases (45%), tumors (45%), and neurological diseases (44%) [20,33]. In Europe, it is estimated that 33 million adults are malnourished or at risk of undernutrition [34]. Malnutrition can affect individuals of any age, from infants to adults and older people, in any setting, either in hospital, in care homes, or at home [35,36,37]. As could be expected, the prevalence is significantly higher in the elderly [38], especially in healthcare settings [39]; in people with long-term conditions, such as gastrointestinal disease, kidney disease, and diabetes, and people with chronic progressive conditions, such as tumors or mental diseases [39]. Malnutrition is a state related to an increased disability, morbidity, short- and long-term mortality, and cost of care [40]. It is well documented that malnutrition is associated with higher complication rate and risk of infection [41,42], increased frailty risk [43], poorer quality of life [44], longer hospital stays, and hospital readmissions [45,46]. For all these reasons, malnutrition represents not only a major clinical and public health problem but also an economic issue. The management cost for a malnourished patient is twice that of a properly fed patient [45,46]. A timely manner of nutritional care, both in terms of prevention and treatment, came to the fore in reducing medical complications, improving the use of healthcare resources and for its potential cost savings. Several nutrition screening tools have emerged in the recent decades. However, there is no consensus on a single methodology capable of assessing a patient’s nutritional status and predicting poor prognosis [47]. Among all, the ESPEN suggests the simultaneous application of the Nutrition Risk Screening-2002 (NRS2002) and the Malnutrition Universal Screening Tool (MUST) [20]. The former was designed to recognize malnourished hospitalized patients likely to benefit from nutritional support, while the latter to forecast the clinical outcome [47]. After the identification of subjects at risk in agreement with the nutritional screening tools mentioned above, the nutritional assessment, through a comprehensive and in-depth evaluation of nutritional status, provides the basis for the diagnosis selection [20]. Medical nutrition aims at supporting patients who, due to a specific disease or medical condition, are temporarily or permanently unable to meet their nutritional and metabolic needs [20]. A spectrum of nutritional support strategies is available, including oral nutritional supplement, enteral nutrition (through the gastrointestinal tract), and parenteral nutrition (intravenous feeding) [20]. Oral nutritional supplements (ONSs) are a type of food for special medical purposes (FSMPs), designed as evidence-based nutritional options for disease-related malnutrition [20,48]. These are intended for patients with a restricted or impaired capability to take, digest, absorb, metabolize, and/or eliminate common foods, specific nutrients, or metabolites, or patients with other nutrient requirements that cannot be fulfilled by sole dietary intervention [20,48]. FSMPs are therefore highly regulated and should be used only on the recommendation of, and under the supervision of, a healthcare professional [48]. Enteral tube feeding is a type of nutrition administered directly into the digestive tract, far-end to the oral cavity [20]. In the absence of contraindications (intestinal failure, bowel obstruction, prolonged paralytic ileus, high-loss intestinal fistulae, mesenteric ischemia, abdominal compartment syndrome, inability to access the gut) [2], enteral nutrition represents the nutrition method of choice in patients who have a functioning digestive system but who are unable to be fed orally [49,50]. Several enteral feeding tubes are disposable and are typically sorted by site of insertion and position of the feeding tube distal tip [50]. The tube could be placed through the nose, i.e., naso-gastric, naso-post pyloric or naso-jejunal tube feeding; or via a stoma that could be placed endoscopically or surgically, i.e., percutaneous endoscopic gastrostomy (PEG) or with a jejunal extension (PEG-J), percutaneous endoscopic jejunostomy (PEJ), gastrostomy, or jejunostomy [20]. Enteral nutrition formulas are classified into four major types: standard (polymeric), semi-elemental (oligomeric), elemental (monomeric), and specialized [51]. Standard formulas contain all the required nutrients, such as proteins, complex carbohydrates, and mainly LCTs (long-chain triglycerides) [52]. Semi-elemental and elemental formulas contain partially or fully hydrolyzed (predigested) components to lighten the digestive system load in treating and absorbing them [51]. Specialized formulas contain biologically active substances or nutrients designed for meeting different nutritional and energy needs in a wide range of clinical conditions or diseases [51,52]. Parenteral nutrition (PN) is a type of nutrition therapy supplied through intravenous administration bypassing the digestive system via a central or peripheral venous line [20]. The need for parenteral nutrition is related to diseases and conditions that impair food intake, digestion, or absorption, including abdominal surgery, short bowel syndrome, gastrointestinal bleeding, obstruction or pseudo-obstruction, enteric-cutaneous fistulas, colitis, inflammatory bowel diseases, or cancer [53,54]. Parenteral nutritional formulas are generally “triple-chamber bags” for all-in-one nutrition solutions. These typically supply dextrose as carbohydrates, essential and nonessential amino acids as proteins, and lipid emulsions for fats, all of which are mixed with electrolytes, trace elements, vitamins, and water [53]. A possible alternative is dual-chamber bag formulas in which only amino acids and glucose are combined, along with electrolytes, vitamins, and trace elements [20]. It is worth noticing that, even in the triple-chamber bags, arginine and glutamine are not included in the parental nutritional formulas currently used, and must therefore be integrated separately [53].

## 5. Nutrition in Mild Acute Pancreatitis

Patients exhibiting mild pancreatitis manifestations, after a short period of fasting, generally can initiate an early solid oral diet and, as such, do not necessitate tailored nutritional interventions [55]. Moreover, amylase and lipase levels should not condition diet advancing [56]. Oral feeding with solids may be reintroduced as an alternative to the commonly accepted approach of initiating nutrition with clear liquids followed by gradual progression to a solid diet. Thus, it has been demonstrated that patients who started eating a full solid diet presented a decreased duration of hospitalization without abdominal pain relapse if compared to patients who received a clear liquid diet [57]. Around 80% of patients can initiate an oral refeeding within 7 days of hospitalization. When not possible, enteral and/or parenteral nutrition is recommended [56].

## 6. Nutrition in Severe Acute Pancreatitis

Adequate nutrition therapy is fundamental in management of severe acute pancreatitis [18]. Therefore, it is essential to prevent malnutrition with a timely diet to avoid depletion of essential nutrients, electrolytes, and disturbances in the balance of acid and base metabolism, which typically occurs with AP [58]. Furthermore, severe acute pancreatitis (SAP) is characterized by a state of hypermetabolism and high protein decomposition. The presence of multiple proinflammatory mediators and resulting systemic inflammatory response syndrome (SIRS) can be attributed to this phenomenon [58]. Nutrition management prevents malnutrition and obviates systemic inflammation, leading to a reduction in pancreatitis complication which helps modify the course of the disease [59,60]. Initially, fasting was considered necessary to guarantee pancreatic rest in order to reduce inflammation through pancreatic stimulation until serum enzyme levels return to normal. Thereafter, parenteral nutritional therapy was deemed as the optimal method of supplying nutrients due to its capability to prevent pancreatic stimulation while furnishing nutritional support [3]. However, parenteral nutrition presents several complications, including the occurrence of infection associated with the central venous catheter or metabolic complications such as hyperglycemia and hyper electrolyte imbalance syndrome [59]. More importantly, parenteral nutrition is not useful to prevent intestinal atrophy [61]. Overgrowth bacteria in the small bowel, alterations in gastrointestinal motility, and heightened permeability of the mucosal barrier are all thought to be responsible for bacterial and endotoxins translocation from the gut to systemic circulation and subsequent pancreatic necrosis superinfection [62]. Due to the above, enteral nutrition has been shown to be more efficacious in mitigating pancreatic necrosis infection in individuals with severe acute pancreatitis compared to total parenteral nutrition [59]. As a matter of fact, enteral nutrition safeguards both the intestinal barrier function and gut microbiological flora and restores intestinal motility [61]. Enteral nutrition is also able to preserve the structural integrity of the gut epithelium by stimulating intestinal contractility and increasing splanchnic blood flow [63]. Besides reducing risk of necrotic infection, total enteral nutrition (TEN) is also associated with lower risk in developing organ failure (21%), if compared with patients on TPN (80%). Patients receiving TPN exhibited a heightened prevalence of infectious necrosis, necessitating surgical intervention in the majority of cases [59]. Hui et al. compared EN with TPN and with enteral plus total parenteral nutrition group (EN + TPN). They highlighted that the EN group of patients had lower rates of incidence of multiple organ dysfunction syndrome than those in TPN and EN + TPN groups [61]. Therefore, it is evident that enteral nutrition exhibits superiority over parenteral nutrition, as it reduces pancreatic necrotic infection rate, multiple organ failure, mortality, and shortens hospital stay [64,65]. Furthermore, Klek et al. revealed that enteral nutrition is linked to supplementary advantages, including a substantially elevated rate of closure and shortened duration of closure of postoperative pancreatic fistulae [66]. In conclusion, international guidelines recommend enteral nutrition as the preferable method over parenteral nutrition for patients with acute pancreatitis, unless contraindications or intolerance to enteral nutrition exist [2].

## 7. Timing of Nutrition in Acute Pancreatitis

The appropriate timing to initiate oral feeding in patients afflicted with acute pancreatitis is still a matter of debate, as it is able to affect the duration of hospital stay and disease outcomes [55]. In mild pancreatitis, oral feeding can be restarted, once abdominal pain diminishes with improvement of inflammatory markers, without waiting for the complete resolution of pain or laboratory abnormalities [67]. The American Gastroenterological Association recommends starting oral feeding within 24 h for patients with mild acute pancreatitis [68]. Enteral nutritional support should be initiated within 24 to 72 h from admission for individuals unable to orally intake food [2]. Gus et al. first described in their meta-analysis the impact and safety of prompt enteral nutrition following admittance in mild acute pancreatitis patients. Immediate EN compared to early refeeding (typically after a brief period of fasting pursued by a gradual intake of food) not only significantly decreases the length of hospital stay but also relieves the intolerance of feeding [69]. The right timing to start nutritional support in severe acute pancreatitis or foreseen severe acute pancreatitis has also been largely debated. Based on the consensus of most authors, EN should start within a range of 24 to 48 h from admission [4]. In addition, the PYTHON trial demonstrated no superiority between early enteral feeding within 24 h through a nasoenteric feeding tube and oral feeding following 72 h in reducing rates of morbidity or mortality associated with acute pancreatitis in individuals at high risk for adverse effects [70]. To clarify, a recent multicenter study pointed out that hospital mortality did not significantly differ from EN within 24 h and EN 24 and 48 h after SAP diagnosis, but was significantly lower if EN was administered within 48 h than over 48 h. The need for surgical intervention was analyzed as a secondary outcome. It emerged that starting enteral nutrition prior to 24 h post-admission leads to greater risk of requiring surgery in comparison to starting enteral nutrition at 24–48 h from admission [71]. Summing up, the timely initiation of enteral nutrition within 48 h from admission shows a significant reduction in mortality, multiple organ failure (MOF), surgery, systemic infections, and localized septic complications in individuals with severe acute pancreatitis (SAP) or predicted SAP, compared to administering enteral nutrition late or parenteral nutrition [72,73].

## 8. Focus on Enteral Nutrition

### 8.1. Route of Administration

From the standpoint of enteral nutrition being superior to parenteral nutrition for pancreatitis, the best route for enteral supplements and the best enteral formula still remain to be established. Historically, nasojejunal tube feeding has been considered the best practice in patients with acute pancreatitis, limiting the use of nasogastric feeding only as an alternative route [74]. The rationale behind this statement can be found in the extra considerations that the pathophysiology of pancreatitis leads to. The employment of jejunal feeding allows to “rest the pancreas”, reducing pancreas stimulation and minimizing its exocrine function [74,75]. More recently, the dogma of “pancreatic rest” is under challenge, with more and more researchers suggesting that nasogastric tube feeding (NGT) is as safe and structured as nasojejunal tube feeding (Table 1). A 2006 meta-analysis from the critical care literature [76] demonstrated that the early use of post-pyloric feeding in lieu of gastric feeding was not correlated with significant clinical advantages in critically ill patients with no evidence of impaired gastric emptying. On a similar note, the works of Kumar [77] and Petrov [78] demonstrate the safety and efficacy of nasogastric (NG) and nasojejunal (NJ) feeding with no pain recurrence or worsening in severe acute pancreatitis (SAP). In 2013, Petrov [78] published the first randomized controlled trial (RCT) to balance NG feeding against a standard nil per os (NPO) regimen in patients with mild to moderate acute pancreatitis. His group proved not only that early commencement (within 24 h of hospital admission) of NG tube feeding was well tolerated, but also that, compared to NPO regimen, it significantly decreased abdominal pain, need for analgesic, and risk of oral food intolerance. Singh et al. [79] demonstrated the noninferiority of NG feeding to NJ feeding, even in patients with SAP. Despite the conclusions of all the aforementioned studies, NJ feeding was still more commonly employed, especially for the NG feeding concerns of increasing likelihood of aspiration pneumonitis [80] and exacerbating acute pancreatitis by stimulating pancreatic secretion [81]. Regarding these concerns, Chang’s work [82] showed no significant difference between post-pyloric and gastric tube feeding with respect to tracheal aspiration and exacerbation of pain as well as energy balance, diarrhea, and mortality rate. These results align with Nally’s evidence on the efficacy and safety of NG nutrition, which was based on a systematic review and meta-analysis including more trials and a larger number of patients [74]. Furthermore, Zhu [83] performed an up-to-date meta-analysis following the newly established criteria (Atlanta 2012 classification) for SAP and reached similar conclusions. On a similar note, Guo et al. stated that NG may be the preferred feeding solution in patients with SAP [61]. Despite all the above, it was only in 2015 that the employment of nasogastric tube feeding was definitively found to have no influence on the patient’s quality of life [84]. Moreover, being a simple bedside procedure not requiring specialized staff, NG tube placement has the potential to allow for an earlier administration of nutrients [74,85,86]. Lastly, NG tube feeding constitutes an economically preferable solution, a nearly indispensable characteristic within today’s conscious healthcare systems [87]. ESPEN guidelines [2], from the standpoint of the aforementioned RCTs [77,79,86] and meta-analyses [74,78,82], recommend the use of an NG tube as a first choice when enteral feeding is required. They also identify the use of the enteral administration via an NJ tube as preferable in case of digestive intolerance [88,89], in the case of patients undergoing minimally invasive necrosectomy who are unable to be fed orally [90,91], and in the case of patients with increased intraabdominal pressure [92]. The ESPEN recommendations align with the UK guidelines [93,94], but are in contrast with the American Gastroenterological Association [68]. The latter, based on a Cochrane meta-analysis [95], recommends using either an NG or NJ tube. This was due to the insufficient quantity and quality of evidence provided arising from the small number of patients and the different criteria and endpoints deployed in the studies. The difference is the case of patients requiring enteral feeding for a prolonged period (>30 days). In such cases, NGT or NJT bring about different hurdles such as discomfort, dislocation, unintentional tube removal, sinusitis, aspiration, and trauma of nasal cavity. According to general nutritional recommendations, percutaneous gastrostomy or microjejunostomy should be considered [96,97]. To sum up, both nasogastric (NG) and nasojejunal (NJ) feeding represent feasible enteral nutrition (EN) solutions for severe acute pancreatitis; however, the existence of an optimal approach is still a topic of debate among experts. There is a need for further studies even if it seems that the NG feeding has the potential to take the place of NJ feeding, according to the circumstances. 

### 8.2. Composition of Enteral Formulas

Another fundamental question relates to the optimal composition of enteral feeding (elemental, semi-elemental, or standard) in acute pancreatitis patients and the possible need for a specialized formula (Table 2). Traditionally, elemental and semi-elemental formulas have been most commonly used [100] for their superior absorption profile, decreased pancreatic stimulation, and better toleration [101,102,103]. These advantages align with the old paradigm whose cornerstone is that the worsening of the acute inflammatory process might be prevented with an attenuation of the pancreatic secretory response [104]. Theoretically, this may be achieved by the employment of a formulation that does not require pancreatic enzymes for digestion (such as elemental or semi-elemental) [105]. At the end of the 1990s, Windsor [106] conducted a prospective study designed to determine whether total enteral nutrition (TEN) with standard formula could reduce the acute response and ameliorate clinical disease severity in patients with acute pancreatitis when compared with total parenteral nutrition (TPN). Proving that enteral feeding regulates the inflammatory response with a consequent better clinical outcome, Windsor, for the first time, suggested the potential use of standard formulations in patients suffering with acute pancreatitis. These conclusions align with those of Gupta’s work [107], but are in contrast with those of Powell [108]. Specifically, the standard formula was compared in the former with parenteral nutrition and in the latter with no nutritional intervention. In 2006, Tiengou et al. [102] presented the first randomized study designed to compare semi-elemental and polymeric formulations in AP patients requiring NJ nutrition. Their work pointed out that, despite both formulations being well tolerated, within the semi-elemental group the infection rate and the median length of hospital stay were found to be shorter. In all the aforementioned studies, as well as in those of Pupelis [109] and Makola [110], enteral nutritional formulas were administered by an NJ tube feeding. However, standard formulas can be administered also by a nasogastric tube feeding, as demonstrated by Eckerwall [85]. A systematic review and meta-analysis of enteral nutrition formulations by Petrov [105] found that the use of cheaper [110] standard formulations does not result in reduced feeding tolerance or in higher infectious complications and mortality rates when compared with semi-elemental or elemental formulas. Moreover, a more recent and larger-scale study in Japan [111] assessed no clinical advantages of the elemental formula in comparison with other formulae in terms of risk of sepsis, hospital-free days, total healthcare costs per admission, and in-hospital mortality. According to the aforementioned studies, there is not sufficient evidence to account for the routine use of an elemental formula in the initiation of enteral feeding for patients with AP. In response to all these conclusions, recent guidelines [2,68,90,112,113] generally recommend the use of a standard polymeric diet.

## 9. The Role of Immunonutrition in AP

In AP, a significant dysregulation of the immune system occurs [4]. As a result, during the acute phase, an enormous release of proinflammatory cytokines occurs, resulting in the systemic inflammatory response (SIRS) [16]. When SIRS is prolonged, we assist in an increased risk of multiorgan failure (MOF) [16]. Inflammation is driven by tumor necrosis factor-alfa (TNF-α), which occurs in the production of proinflammatory cytokines such as interleukin-1 (IL1), interleukin-6 (IL6), interleukin-8 (IL8), and Interferon-gamma (IFN-γ). Furthermore, these cytokines are implicated in the recruitment of neutrophils, T-cells, and macrophages [114]. Cytokines may represent an early predictor of the gravity of AP [115]. Numerous studies have proven that the level of IL-6 in the plasma is a sensitive marker for predicting organ failure and SAP [116]. Concurrently, there are contradictory opinions over the role of TNF-α as a prognostic factor in SAP [117]. While some authors have argued that high levels of TNF-α can predict SAP, other authors showed that in only a tiny percentage of patients with SAP was a high level of this cytokine present [118]. As is known, the handling of AP depends on how severe the disease is [4]. Thus, in recent years, clinicians centered their attention on the possible immunomodulatory function of nutrition, giving rise to the concept of “immunonutrition” [119]. Immunonutrition makes use of elements called “immunonutrients” that modify an individual’s immune system and inflammatory response [120]. Over the years, several RCTs [121,122,123,124] and meta-analyses emphasized the advantageous effect of immunonutrition [119,125]. Zhou et al., in their work comprising 14 articles and 568 total subjects, showed that immunonutrition is associated with an improved mortality, decreased infections rate, and shorter hospitalization time [119]. Moreover, other studies have shown that there are no significant advantages in the enteral administration of immunonutrients [126,127]. In particular, Petrov et al. proved in their meta-analysis that immunonutrient supplementation was not beneficial in terms of total infectious complications and mortality with respect to standard enteral nutrition [127]. Further to this, Poropat et al., in their work including 15 studies, also found insufficient or meagre-quality evidence about the safety and the efficacy of immunonutrition [126]. ESPEN does not recommend the routine use of immunonutrients in SAP, but when EN is not possible, parenterally administered glutamine is the only immunonutrient recommended [2].

## 10. The Immunonutrients

Glutamine (Gln), omega- 3-unsaturated fat acids (PUFAs), arginine (Arg), and nucleotides are the most noted immunonutrients [4]. Glutamine is an essential amino acid, useful for the regular function of gut-associated lymphoid tissue (GALT), and it is implicated in the glutathione’s synthesis, an important antioxidant [4,128]. Glutamine influences nitric oxide metabolism, lymphocyte and monocyte activity, regulates cell maturation, and stimulates heat shock proteins (HSPs) production [128,129]. In addition to its immunomodulatory role, it also acts directly on the gut barrier. Huang et al. showed that enteral administration of glutamine increases intestinal barrier function by reducing intestinal permeability in early AP stages [124]. Furthermore, a 2016 meta-analysis, including ten works, demonstrated that the nutrition support of glutamine increases the level of albumin, decreases C-reaction protein level, and reduces the number of infectious complications, especially if parenterally administered [130]. In addition, another meta-analysis of 505 patients and 12 RCTs showed that intravenous administration of glutamine reduces risk of mortality, but not length of hospital stay [121]. Finally, glutamine has been shown to have positive effects in preventing complications [131]. ESPEN guidelines recommended L-glutamine at a dosage of 0.20 g/kg per day [2]; however, in critical patients, the role of glutamine is still controversial. Several studies have demonstrated that the administration of glutamine in this type of patient is not associated with any mortality rate and LOS benefit [132,133,134].

Arginine (Arg) is a semi-essential amino acid essential in protein synthesis. Therefore, it may stimulate tissue growth after trauma [135,136], cause T-cell activation and reduce the production of proinflammatory mediators such as TNF alpha, IL6, and IL18, as also demonstrated by Yeh et al. [137,138]. Nonetheless, arginine constitutes a substrate for nitric oxide (NO); excessive NO production can result in systemic hypotension and tissue damage [128]. For this rationale, caution should be used when administering arginine, especially in patients in hyperinflammatory states.

Omega 3-unsatturated fatty acids (PUFAs) have anti-inflammatory action. A randomized and controlled study showed that adding PUFAs to parenteral nutrition decreases inflammatory response, acting on eicosapentaenoic acid (EPA) concentration and on proinflammatory cytokine levels [139]. In addition, Lei et al. demonstrated that intravenous PUFA supplementation is associated with reduced hospitalization time, lower mortality, and decreased rates of infectious complications [140].

Therefore, the role of immunonutrients in managing SAP support is still controversial and requires further investigation to be defined.

## 11. The Role of Gut Microbiota in AP

It is well known that the gut microbiota is involved in maintaining health and modulating the immune response of an individual [141]. The available data in the literature emphasize the crucial role of the microbiota in the pathogenesis of various diseases, such as inflammatory bowel disease (IBD) [142], diabetes mellitus [143], colon rectal cancer [144], and neurological disorders [145], including pancreatic disorders [146]. Gut microbiota is mainly composed of 80–90% bacteria belonging to Firmicutes and Bacteroides phyla, other bacteria belonging to minor phyla, such as Actinobacteria, Fusobacteria, Proteobacteria, and Verrucomicrobia, and different families of fungi [147]. The gut microbiota’s altered composition is due to intestinal barrier dysfunction [148]. Various evidence supports the involvement of microbiota and intestinal dysbiosis also in pancreatic diseases, including AP [149]. Disorders in microcirculation, hypovolemia, and subsequent intestinal ischemia that occur during AP can damage the intestinal barrier, resulting in barrier dysfunction in 59% of patients [150,151]. As a result, intestinal bacteria can translocate into the systemic circulation and pancreatic environment, normally sterile in a healthy individual, and this influences the progression of AP [150,152]. Moreover, some authors support that changes in microbiota could foretell the severity of AP [153]. In patients with infected pancreatic necrosis, mortality and risk of multiorgan failure are twice as high than those of patients with a sterile pancreatic environment [154]. In addition, thanks to the widespread use of antibiotic prophylaxis, the main flora that infects pancreatic necrosis is composed of Enterococcus, Staphylococcus, and fungi, as well as Candida [155]. In any case, the composition of the microbial spectrum does not influence mortality [156]. Furthermore, as demonstrated in the murine model, we assist in reducing antimicrobial peptides (AMPs) during AP [157,158]. AMPs are peptides mainly secreted by intestinal Paneth cells but also by acinar cells of the pancreas, which are involved in innate immunity and contribute to maintaining the gut environment [159]. The lower level of AMPs may facilitate bacteria overgrowth, leading to proinflammatory and systemic response [158]. The gut is not only passively involved in AP, but is a crucial character in the severity of the disease [150]. Particularly, Li et al. suggested that dysbiosis and the consequent reduction in short-chain fatty acids (SCFAs) production are involved in AP progression [160]. SCFAs are microbiota metabolites, and they stimulate the release of cathelicidin-related antimicrobial peptides (CRAMP) by beta-cells, thereby acting on the gut microenvironment [161]. Conversely, as previously explained, AMPs regulate the composition of the intestinal microbiota [158]. Therefore, one pancreas–gut axis might be hypothesized (Figure 1) [162].

In AP, we observe a lowering in Bacteroides and Lactobacillus and an increase in Enterobacteriaceae and Firmicutes [153]. In addition, Zhu et al. showed an association between the AP severity and the increase in opportunistic bacteria, such as Escherichia-Shigella, and the reduction in the abundance of beneficial bacteria, such as Blautia [163]. Even though evidence supporting the role of gut dysbiosis is still scarce, we believe maintaining the integrity of the gut barrier and microbiota are key elements in treating AP. To this end, several studies and trials have focused on modulating microbiota in AP, as a treatment strategy, but evidence in the literature has yet to be uncovered [164,165,166,167].

## 12. Modulation of Microbiota in AP

The prophylactic use of antibiotics in PA, defined as the use of antibiotics in the absence of clinical infection and to prevent pancreatic infection, does not significantly reduce mortality and morbidity [18]. A 2009 meta-analysis including eight RCTs and 502 patients, randomized to receive antibiotics prophylaxis (n = 253) or placebo (n = 249), showed no improvement in mortality or no protection against infection [168]. On the same side, another meta-analysis demonstrated in the antibiotic-treated group a nonsignificant reduction in mortality (8.4% vs. controls 14.45%) and in the rate of infected necrosis (19.7% vs. controls 24.4%). The only exception was imipenem, taken alone, for which a significant reduction in the infection was observed [169]. More recently, Guo et al. conducted a meta-analysis including 3864 patients with SAP to evaluate the efficacy of antibiotic prophylaxis with carbapenem in patients with SAP [170]. The administration of carbapenem as antibiotic prophylaxis in SAP was associated with a statistically significant reduction in complications (OR = 0.48; *p* = 0.009) and infections (OR = 0.27; *p* = 0.03). However, despite these results, no statistical difference was observed in the various subanalyses in terms of mortality, incidence of infected pancreatic necrosis, and need for intensive treatment. Consequently, in the absence of strong supporting evidence, the authors do not recommend the routine use of carbapenem antibiotics [170]. According to actual guidelines, antibiotic prophylaxis in AP is not recommended [18,67,171], and additional studies are required to establish its role.

Over the years, several works [126,164,172] have focused on the role of probiotics in AP, with inconsistent and sometimes conflicting results, especially regarding safety, adverse effects, and reduction of infection and mortality rates. Likewise, prebiotics, previously defined as undigestible substances that influence the microbiota’s activity or composition, are now defined as substrates beneficial to health used by host microorganisms [173]. Olàh et al., in an RCT that included 45 patients divided into a treatment group, receiving a freeze-dried preparation of live probiotics with oat fiber, and a control group, receiving an inactivated form, showed that the addition of L. plantarum 299 was shown to be effective in decreasing sepsis and surgery [165]. Similarly, the same study group in 2007 evidenced that the early enteral administration of a symbiotic (a combination of probiotics and prebiotics), named Synbiotic 2000, in the late phase of SAP can reduce the incidence of SIRS, infections, and mortality rate [174]. In the PROPATRIA study [166], a multicenter randomized, double-blind, placebo-controlled trial, 298 patients were randomized to receive a probiotics formulation composed of two different species of Bifidobacterium, three species of Lactobacillus, and one of Lactococcus or placebo. The probiotic group showed a higher rate of mortality (16% vs. 6%) and intestinal ischaemia (6% vs. 0%) [166]. These results were initially attributed to the probiotic mixture used. A few years later, Bongearts et al. reconsidered this study, suggesting that a fatal combination of pancreatic proteolytic enzymes and probiotics caused the high mortality rate in the PROPATRIA study. Ultimately, authors suggested that probiotics can be used in AP, provided it is started immediately after the onset of symptoms and with caution to prevent the overgrowth of bacteria [175]. Wan et al. conducted an RCT to investigate the effects of a combination of Bacillus subtilis and Enterococcus faeciumin on reducing LOS in patients with mild disease [176]. Their work showed a relevant reduction in LOS in the probiotic group [176]. In 2022, an RCT evaluated the efficacy of synbiotics in reducing infective complications in moderately severe and severe AP [177]. The results showed no significant differences in septic complications between the groups (59% vs. 64%) but there was a significant reduction in the duration of hospitalization (10 vs. 7) [177].

Several trials have shown that soluble dietary fibers (SDFs) impact gut integrity and regulate gut microbiota [178]. Chen et al. carried out a single-blind randomized controlled study underlining the potential of SDF in subjects with SAP [179], including forty-nine patients. The study group randomly received, in addition to enteral nutrition solution, 20 g/day of polydextrose. The administration of polydextrose demonstrated increased intestinal motility and gastrointestinal hormone production, reducing the rate of feeding intolerance in the study group (59.09% vs. 25.00%, *p* < 0.05) [179].

The contradictory results emerging from the literature on the use of probiotics are probably due to the extreme heterogeneity in dose, type, and timing of administration, as also argued by Gou et al. [164]. Thus, we do not have uniform data in the literature. Therefore, further studies that are better designed are needed to better define the safety and the efficacy of probiotics in AP management.

## 13. Conclusions

To sum up, in patients with predicted mild AP, oral feeding should be started as soon as possible regardless of serum lipase and amylase levels. In the event of oral feeding intolerance, enteral nutrition should be administered within 24–74 h to keep gut mucosa integrity and intestinal motility. In patients suffering from severe AP who cannot be fed orally, enteral nutrition should be introduced within 48 h from hospital admission. Parenteral nutrition, instead, should be administered only in patients who are unable to tolerate EN or have contraindications for it. From the standpoint of enteral nutrition being superior to parenteral nutrition for pancreatitis, ESPEN guidelines recommend NG tube feeding as the best route for enteral supplements when enteral feeding is required, relegating the use of NJ tube feeding only to specific clinical settings. However, the existence of an optimal approach is still a matter of debate and further studies are needed. Most authors, instead, agree on the optimal composition of enteral feeding. Therefore, the latest guidelines recommend the use of a standard polymeric diet, thus overcoming the old paradigm of “pancreatic rest”. Glutamine is the only immunonutrient recommended, but the role of immunonutrition and probiotics in AP, to date, remains under discussion.

## Figures and Tables

**Figure 1 nutrients-15-01939-f001:**
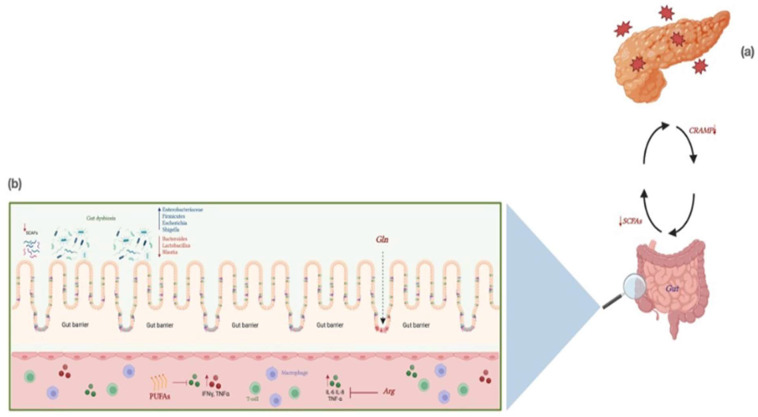
Gut–pancreas axis during AP. (**a**) During AP, we observe a reduction in AMPs production by the pancreas. This facilitates bacterial overgrowth and the release of inflammatory cytokines such as TNF-α, IL-1, IL-6, IL-8, and IFN-γ. At the same time, intestinal dysbiosis is associated with a reduction in the production of SCFAs, resulting in increased intestinal permeability and worsening pancreatic damage. (**b**) Immunonutrients modify the inflammatory response that occurs during AP, and act on the damaged intestinal barrier. SCFAs: short-chain fatty acids; AMPs: antimicrobial peptides; TNF-α: tumor necrosis factor-α; IL-1: interleukin-1; IL-6: interleukin-6; IL-8: interleukin-8; IFN-γ; Interferon-gamma; Arg: arginine; Gln: glutamine; PUFAs: omega-3-unsatured fat acids. Created with BioRender.com.

**Table 1 nutrients-15-01939-t001:** Evidence regarding NG feeding in acute pancreatitis.

First Author	Year	Study Design	Remarks
Kwok M. Ho [76]	2006	RCT	In critically ill adults with a good gastric emptying function, the use of NJ feeding instead of NG feeding was not associated with significant clinical advantages.
Kumar A. [77]	2006	RCT	EN by both NK and NG is well tolerated in patients with SAP without leading to recurrence or worsening of pain.
Petrov MS. [78]	2008	SR	NG route is safe and well tolerated in patients with predicted SAP.
Petrov MS. [98]	2013	RCT	Comparing NG feeding with NPO, the former significantly reduces the intensity and span of abdominal pain, need for analgesic, and risk of oral food intolerance.
Sing N. [79]	2012	RCT	Early NG feeding was not inferior to NJ in patients with SAP.
Chang Ys. [82]	2013	MA	No significant differences were found between NG and NJ tube regarding tracheal aspiration, energy balance, diarrhea, and mortality rate.
Nally D. [74]	2014	SR and MA	NG feeding is fruitful in patients with severe AP.
Zhu Y. [83]	2016	MA	Comparing NG or NJ nutrition in patients with SAP, no significant dissimilarities were found in the mortality rate, infectious and/or digestive complications, achieving energy balance, or length of hospital stay.
Guo Y. [99]	2016	SR and MA	NG may be the feeding solution choice in patients with SAP.
Pendharkar SA. [84]	2014	SR and MA	Nasogastric tube feeding was found to have no influence on the patient’s quality of life.
Eatock FC. [86]	2005	RT	The simpler, inexpensive, and more manageable used NG feeding is as good as NJ feeding in SAP.
Hauschild TB. [87]	2012	CS	NG tube feeding constitutes an economically preferable solution.

CS: comparative study, MA: meta-analysis, NG: nasogastric, NJ: nasojejunal, RCT: randomized controlled trial, RT: randomized trial, SAP: severe acute pancreatitis, SR: systematic review.

**Table 2 nutrients-15-01939-t002:** Evidence regarding the optimal composition of enteral feeding (elemental, semi-elemental, or standard in acute pancreatitis.

First Author	Year	Study Design	Type of Nutrition and Formula	Remarks
Windsor Ac. [106]	1998	CT	Standard formula vs. total parenteral nutrition.	Proving that enteral feeding modulates the inflammatory response with a consequent better clinical outcome, Windsor, for the first time, suggested the potential use of standard formulations in patients suffering from AP.
Gupta R. [107]	2003	RCT	Standard formula vs. parenteral nutrition	Early use of nutritional support in the form of TEN is safe in predicted SAP.
Powell JJ. [108]	2000	RCT	Standard formula vs. no nutritional interventions	Early enteral nutrition was found to have no effect on inflammatory response markers or on organ dysfunction.
Tiengou LE. [102]	2006	RCT	Standard formula vs. polymeric formula by NJ tube feeding.	Despite both formulations being well tolerated, within the semi-elemental group the infection rate and the median LOS were found to be shorter.
Pupelis G. [109]	2001	RCT	Standard formula by NJ tube feeding,	Standard formula by jejunal feeding, even when started late, improves outcomes in patients with SAP.
Makola D. [110]	2006	CT	Standard formula by NJ tube feeding.	Standard enteral formula is effective in the management of patients with complicated AP.
Eckerwal GE. [85]	2006	RCT	Standard formula by NG tube feeding vs. total parenteral nutrition	Standard formula can also be administered by NG route.
Petrov MS. [105]	2009	SR and MA	Standard formula vs. semi-elemental formula.	The use of polymeric formulation is not associated with a significantly higher risk of feeding intolerance, infectious complications, or mortality rate.
Endo A. [111]	2008	RCS	Elemental formula vs. semi-elemental formula and standard formula.	No clinical advantages of the elemental formula in comparison with other formulae in terms of risk of sepsis, hospital-free days, total healthcare costs, and in-hospital mortality.

CT: clinical trial, MA: meta-analysis, NG: nasogastric, NJ: nasojejunal, RCT: randomized controlled trial, RCS: retrospective cohort study, SAP: severe acute pancreatitis, SR: systematic review.

## Data Availability

Not applicable.

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
