# Peer review of "Nutrition in Acute Pancreatitis: From the Old Paradigm to the New Evidence"

_nutrients, 2023, doi:10.3390/nu15081939_

Round 1

Reviewer 1 Report

Generally, the subject of this paper is noteworthy and has practical aspects. The aim of the work was to collect and analyze evidence of the nutritional aspects of acute pancreatitis (AP). The work has been written in an accessible language, it touches on numerous aspects related to nutrition in AP, both in the past and present. The authors thoroughly analyzed the available publications, presented them in their study, and drew the right conclusions. The publication contains thirteen subchapters, each dealing with a different issue related to nutrition in AP. The chapters are arranged in a logical order except for one on the role of immunonutrition and AP, which should be placed after the chapter “The immunonutrients”.

The following points required explanation and revision:

-in subchapter 3. Acute pancreatitis, one sentence is repeated, but there is another citation at the end - which is the correct one? maybe both? (9, 10)

-in subchapter 4. Clinical Nutrition,  the abbreviation FFMI has not been explained

-in subchapter 12. Modulation of Microbiota in Acute Pancreatitis there are sentences that in the meta-analysis included a total of 3.864 SAP patients the administration of carbapenem as antibiotic prophylaxis in SAP was associated with a statistically significant reduction of complications and infections. The next sentence: Routinary use of carbapenem antibiotics is not recommended (item 173 in references). The authors should explain why they are not recommended despite the reduction of complications and infections.

I believe that research paper submitted for referring, after completing all aforementioned small corrections, may be published in Nutrients.

Author Response

Many thanks for giving us the opportunity to resubmit our paper “NUTRITION IN ACUTE PANCREATITIS: FROM THE OLD PARADIGM TO THE NEW EVIDENCE ”.

We also wish to thank the reviewers for their time and effort. We went through their interesting suggestions that allowed us to improve our paper.

Please, find a point-by-point answer to all addressed issues:

REVIEWER 1

Generally, the subject of this paper is noteworthy and has practical aspects. The aim of the work was to collect and analyze evidence of the nutritional aspects of acute pancreatitis (AP). The work has been written in an accessible language, it touches on numerous aspects related to nutrition in AP, both in the past and present. The authors thoroughly analyzed the available publications, presented them in their study, and drew the right conclusions. The publication contains thirteen subchapters, each dealing with a different issue related to nutrition in AP. The chapters are arranged in a logical order except for one on the role of immunonutrition and AP, which should be placed after the chapter “The immunonutrients”.

  1. We thank the reviewer for the interesting point of view. However, this order was purposely provided in order to give a more fluid reading and a better understanding of the concept of “immunonutrition” by first describing its role and secondly providing the examples of “immunonutrients” able to influence gut barrier.

The following points required explanation and revision: 

  1. in subchapter 3. Acute pancreatitis, one sentence is repeated, but there is another citation at the end - which is the correct one? maybe both? (9, 10). R: We thank the reviewer for having pointed out this discrepancy. The correction was made with the insertion of the appropriate citation.

  1. in subchapter 4. Clinical Nutrition,the abbreviation FFMI has not been explained.

R: We thank the reviewer for this comment. The abbreviation "FFMI" has now explicated in its full name: “Fat Free Body Mass Index”.

  1. in subchapter 12. Modulation of Microbiota in Acute Pancreatitis there are sentences that in the meta-analysis included a total of 3.864 SAP patients the administration of carbapenem as antibiotic prophylaxis in SAP was associated with a statistically significant reduction of complications and infections. The next sentence: Routinary use of carbapenem antibiotics is not recommended (item 173 in references). The authors should explain why they are not recommended despite the reduction of complications and infections.

R: We thank the reviewer for the comment. As suggested, we modified subchapter 12, to better explain why routine use of carbapenem antibiotics is not recommended despite the reduction in infections and complication.

Reviewer 2 Report

Manuscript ID: nutrients-2335373

Title: Nutrition in acute pancreatitis: from the old paradigm to the new evidence

Authors Sara Sofia De Lucia et al. 

The manuscript is a very interesting and well-written. The authors present the history and current state in nutritional management of acute pancreatitis. In addition, the authors conducted the analysis of several still-discussed topics in their manuscript to provide a comprehensive overview of nutritional management in patients with acute pancreatitis.

On the other hand, it should be said that there are some shortcomings and errors in the manuscript that need to be corrected:

  1. Page 2, section 3. Acute pancreatitis, line 6. Replace “100.000” with “100,000”. In English, a period is used to separate whole numbers from decimals.
  2. Page 2, section 3. Acute pancreatitis, lines 6-7. What does mean “with any statistically significant difference among men and women”?
  3. Page 2, section 3. Acute pancreatitis, lines 7-10. The sentence “Xiao et al. in their systematic review…”in an exact repetition of the previous sentence. In addition, the authors present the same sentence twice but support it with different references.
  4. Authors should use uniform spelling. They seem to use mostly American spelling, but there are also words like “hypercalcaemia” (page 2, section 3. Acute pancreatitis, line 6).
  5. All abbreviations should be presented in their full name at the point where they appear for the first time, starting from the abstract. Full names of abbreviation should be repeated in, the body of the manuscript at the place of the first use. This recommendation was generally followed by the authors. However, some abbreviations are used without presenting their full names. For example: ESPEN (page 3, section 4. Clinical Nutrition, line 14), BMI (page 3, section 4. Clinical Nutrition, line 15), FFMI (page 3, section 4. Clinical Nutrition, line 16), ASPEN (page 3, section 4. Clinical Nutrition, line 16), LCTs (page 5, line 1). In the case “malnutrition can be disease related (DRM)” (page 3, section 4. Clinical Nutrition, line 19), the authors should also include the full name of the abbreviation in parenthesis. In addition, in captions under the tables, the authors provided the full names of the abbreviations used, but the abbreviation used in the table headings should also be given in expanded form in these headings. Authors should check the correctness of abbreviation throughout the manuscript.
  6. According to the policy of MDPI, the authors must cite the full title of the paper, page range or article number, and digital object identifier (DOI) where available. References to books should cite the author(s), title, publisher, publisher location (city and country), publication year, and page: The authors should check all references and at least references 22 (number of page, check PubMed), 32 (Where was this article published?), 34 (check the names of authors and name of journal in journal), 35, 36, 43, 46, 125 and 126 should be corrected.

Author Response

The manuscript is a very interesting and well-written. The authors present the history and current state in nutritional management of acute pancreatitis. In addition, the authors conducted the analysis of several still-discussed topics in their manuscript to provide a comprehensive overview of nutritional management in patients with acute pancreatitis.

On the other hand, it should be said that there are some shortcomings and errors in the manuscript that need to be corrected:

  1. Page 2, section 3. Acute pancreatitis, line 6. Replace “100.000” with “100,000”. In English, a period is used to separate whole numbers from decimals.

R: We thank the reviewer for the comment, the text has been modified as requested.

  1. Page 2, section 3. Acute pancreatitis, lines 6-7. What does mean “with any statistically significant difference among men and women”?

R: We thank the reviewer for the comment. We clarified in the section that there is no difference in the incidence of acute pancreatitis between the two genders.

  1. Page 2, section 3. Acute pancreatitis, lines 7-10. The sentence “Xiao et al. in their systematic review…”in an exact repetition of the previous sentence. In addition, the authors present the same sentence twice but support it with different references.

R: We thank the reviewer for the comment. We corrected this point with the appropriate citation.

  1. Authors should use uniform spelling. They seem to use mostly American spelling, but there are also words like “hypercalcaemia” (page 2, section 3. Acute pancreatitis, line 6).

 R: We thank the reviewer for the comment. We corrected “hypercalcaemia” with “hypercalcemia” and uniformed the spelling within the text as requested.

  1. All abbreviations should be presented in their full name at the point where they appear for the first time, starting from the abstract. Full names of abbreviation should be repeated in, the body of the manuscript at the place of the first use. This recommendation was generally followed by the authors. However, some abbreviations are used without presenting their full names. For example: ESPEN (page 3, section 4. Clinical Nutrition, line 14), BMI (page 3, section 4. Clinical Nutrition, line 15), FFMI (page 3, section 4. Clinical Nutrition, line 16), ASPEN (page 3, section 4. Clinical Nutrition, line 16), LCTs (page 5, line 1). In the case “malnutrition can be disease related (DRM)” (page 3, section 4. Clinical Nutrition, line 19), the authors should also include the full name of the abbreviation in parenthesis. In addition, in captions under the tables, the authors provided the full names of the abbreviations used, but the abbreviation used in the table headings should also be given in expanded form in these headings. Authors should check the correctness of abbreviation throughout the manuscript.

R: We thank the reviewer for the comment. We defined all the abbreviations signaled above with their full name: ESPEN (European Society for Clinical Nutrition and Metabolism), BMI (Body Mass Index), FFMI (Fat-Free Body Mass Index), ASPEN (American Society for Parenteral and Enteral Nutrition), LCTs (Long Chain Triglycerides), DRM (Disease-Related Malnutrition).

  1. According to the policy of MDPI, the authors must cite the full title of the paper, page range or article number, and digital object identifier (DOI) where available. References to books should cite the author(s), title, publisher, publisher location (city and country), publication year, and page: The authors should check all references and at least references 22 (number of page, check PubMed), 32 (Where was this article published?), 34 (check the names of authors and name of journal in journal), 35, 36, 43, 46, 125 and 126 should be corrected.

R: We thank the reviewer for the comment. We checked the references throughout the manuscript and made the following correction: added the number of pages for reference 22, specified that reference 32 was published on BAPEN (British Association for Parenteral and Enteral Nutrition), modified the names of authors and name of the journal for reference 34, and corrected reference 35, 36, 43, 46, 125 and 126 as indicated.

Reviewer 3 Report

Dear Editor

 The authors provide a detailed review of nutrition in pancreatitis. The followings are my comment. 

#1. Page 9, Table , Ref 84, provide abbrevation for "RT" in the table

#2. Page 11, Table, Ref 11, "/" should be "RCS" ?

Author Response

 The authors provide a detailed review of nutrition in pancreatitis. The followings are my comment. 

  1. Page 9, Table , Ref 84, provide abbrevation for "RT" in the table.

R: We thank the reviewer for the comment and provided the abbreviation “RT” for Randomized Trial.

  1. Page 11, Table, Ref 11, "/" should be "RCS" ?

R: We thank the reviewer for the clarification. We added the type of study: Randomized controlled trial.